# Sulfonylureas may be useful for glycemic management in patients with diabetes and liver cirrhosis

Fu-Shun Yen[1], Jung-Nien Lai[2,3], James Cheng-Chung Wei[4,5,6], Lu-Ting Chiu[7,8], Chii-Min Hwu[9,10], Ming-Chih Hou[11]*, Chih-Cheng Hsu[12,13,14]*

1 Dr. Yen's Clinic, Taoyuan, Taiwan, 2 School of Chinese Medicine, College of Chinese Medicine, China Medical University, Taichung, Taiwan, 3 Department of Chinese Medicine, China Medical University Hospital, Taichung, Taiwan, 4 Department of Allergy, Immunology & Rheumatology, Chung Shan Medical University Hospital, Taichung City, Taiwan, 5 Institute of Medicine, College of Medicine, Chung Shan Medical University, Taichung City, Taiwan, 6 Graduate Institute of Integrated Medicine, China Medical University, Taichung City, Taiwan, 7 Management Office for Health Data, China Medical University Hospital, Taichung, Taiwan, 8 College of Medicine, China Medical University, Taichung City, Taiwan, 9 Faculty of Medicine, National Yang-Ming University School of Medicine, Taipei, Taiwan, 10 Section of Endocrinology and Metabolism, Department of Medicine, Taipei Veterans General Hospital, Taipei, Taiwan, 11 Division of Gastroenterology and Hepatology, Department of Medicine, Taipei Veterans General Hospital, Taipei, Taiwan, 12 Institute of Population Health Sciences, National Health Research Institutes, Miaoli County, Taiwan, 13 Department of Health Services Administration, China Medical University, Taichung, Taiwan, 14 Department of Family Medicine, Min-Sheng General Hospital, Taoyuan, Taiwan

* mchou@vghtpe.gov.tw (MCH); cch@nhri.edu.tw (CCH)

**Data Availability Statement:** Data are available from the National Health Insurance Research Database (NHIRD) published by Taiwan National Health Insurance (NHI) Bureau. The data utilized in

## Abstract

This study aimed to investigate the long-term outcomes of sulfonylurea (SU) use in patients with T2DM and compensated liver cirrhosis. From January 1, 2000, to December 31, 2012, we selected the data of 3781 propensity-score-matched SU users and nonusers from Taiwan's National Health Insurance Research Database. The mean follow-up time for this study was 5.74 years. Cox proportional hazards models with robust sandwich standard error estimates were used to compare the risks of main outcomes between SU users and nonusers. The incidence of mortality during follow-up was 3.24 and 4.09 per 100 person-years for SU users and nonusers, respectively. The adjusted hazard ratios and 95% confidence intervals for all-cause mortality, major cardiovascular events, and decompensated cirrhosis in SU users relative to SU nonusers were 0.79 (0.71–0.88), 0.69 (0.61–0.80), and 0.82 (0.66–1.03), respectively. The SU-associated lower risks of death and cardiovascular events seemed to have a dose–response trend. This population-based cohort study demonstrated that SU use was associated with lower risks of death and major cardiovascular events compared with SU non-use in patients with T2DM and compensated liver cirrhosis. SUs may be useful for glycemic management for patients with liver cirrhosis.

## Introduction

Type 2 diabetes mellitus (T2DM) may result from insulin resistance or insufficient insulin secretion, which causes inefficient absorption of ingested carbohydrates by the skeletal muscle or liver, resulting in increased blood glucose levels [1]. The number of people with T2DM and

this study cannot be made available in the paper, the supplemental files, or in a public repository due to the "Personal Information Protection Act" executed by Taiwan's government, starting from 2012. Requests for data can be sent as a formal proposal to the NHIRD (http://nhird.nhri.org.tw) or by email to nhird@nhri.org.tw.

**Funding:** This study is supported in part by Taiwan's Ministry of Health and Welfare Clinical Trial Center (MOHW109-TDU-B-212-114004), Ministry of Science and Technology Clinical Trial Consortium for Stroke (MOST 108-2321-B-039-003), Tseng-Lien Lin Foundation, Taichung, Taiwan (URL of each funder website: https://www.taiwanclinicaltrials.tw/ctc; http://www.ltl-charity.org.tw/about.php?id=4). These funding agencies did not influence the study design, data collection and analysis, decision to publish, or manuscript preparation. The writing and preparation of this paper was not funded by any organization, and employees of funders or any author who received the funding did not undertake data analysis. The funders did not offer support for writing. The corresponding authors had complete access to all data in the study and final responsibility for the publication decision. The funders had no role in study design, data collection and analysis, decision to publish, or preparation of the manuscript.

**Competing interests:** The authors have declared that no competing interests exist.

metabolic syndromes has increased considerably because of high-calorie diets and sedentary lifestyles observed globally [2]. Most people with T2DM also have nonalcoholic fatty liver disease (approximately 40%–70%), nonalcoholic steatohepatitis, or even cirrhosis [3].

Patients with liver cirrhosis—which may be engendered by a low liver mass and low extraction of insulin or shunting of circulating insulin from the portal system to systemic circulation—usually exhibit hyperinsulinemia and insulin resistance [4]. Approximately 96% of such patients have glucose intolerance, with 30% of them being diagnosed as having diabetes mellitus [5]. However, medications for managing diabetes in patients with liver cirrhosis are associated with concerns. Insulin and sulfonylurea may lead to hypoglycemia, and metformin may be associated with the risk of lactic acidosis. Furthermore, information about the effectiveness and safety of thiazolidinediones and dipeptidyl peptidase-4 (DPP-4) inhibitors for managing diabetes in patients with liver cirrhosis is insufficient [4]. Accordingly, no consensus is available regarding the optimal glycemic management strategy for patients with liver cirrhosis.

Sulfonylureas (SUs), the first available oral anti diabetic drugs since the 1950s, have been the major antidiabetic therapy for years and are still widely used worldwide because they are inexpensive [6]. SUs bind to the specific sulfonylurea receptors of pancreatic β cells to inhibit $K_{ATP}$ channels and stimulate insulin secretion in a glucose-dependent manner [6]. Research revealed that SUs could reduce blood hemoglobin A1C by nearly 1%–1.5% [6]. Patients usually exhibit favorable initial response to SUs, with the annual secondary failure rate of SUs being 5%–7%. However, most patients may need to be prescribed additional antidiabetic drugs after 10 years [1]. The main side effects of SUs are hypoglycemia and weight gain; nevertheless, the newer generations of SU are less associated with hypoglycemia [7]. Few studies have investigated the use of SUs in patients with liver cirrhosis. Accordingly, to address this research gap, we performed this retrospective cohort study to evaluate the long-term outcomes of SU use in patients with T2DM and compensated liver cirrhosis.

## Materials and methods

### Data source

The dataset we used in the present study was Longitudinal Cohort of Diabetes Patients (LHDB). The LHDB comprises the data of 1,700,000 randomly selected patients with T2DM with longitudinally linked data available from 1997 to 2013. It is a subset of the National Health Insurance Research Database (NHIRD). The NHIRD comprises the health records of people insured in Taiwan's National Health Insurance (NHI)program, which was established in 1995 and covered approximately 99% of Taiwan's 23 million people in 2000 [8]. This administrative dataset includes information about sex, age, disease management, and diagnoses according to the International Classification of Diseases, Ninth Revision, Clinical Modification (ICD-9-CM) codes. We confirmed that all methods were performed in accordance to Declaration of Helsinki. To protect individual privacy, all patient or caregiver data were scrambled before being released. This study was approved by the Research Ethics Committee of China Medical University and Hospital (CMUH104-REC2-115-CR-4) and was exempted from informed consent requirements.

### Participants

We collected data about people newly diagnosed as having T2DM and liver cirrhosis between January 1, 2000, and December 31, 2012, and followed them until December 31, 2013. Patients with recorded diagnoses (ICD-9-CM code 250.xx) in at least two outpatient visits within 1 year or one admission with the prescription of hypoglycemic agents were defined as having T2DM. Patients with recorded diagnoses (571.5, 571.2, or 571.6) in at least two outpatient visits within 1 year or one admission were defined as having liver cirrhosis. Previous studies have

validated the use of ICD-9-CM codes to define T2DM and liver cirrhosis [9, 10]. Patients with liver cirrhosis and bleeding esophageal varices (456.0 or 456.2), ascites (789.59 or 789.5), hepatic encephalopathy (572.2), or jaundice (782.4) were defined as having decompensated liver cirrhosis [11] and were initially excluded from this study. Patients without these cirrhotic complications were defined as having compensated liver cirrhosis. We excluded patients who were diagnosed as having type 1 diabetes mellitus (250.1); were younger than 18 years or older than 80 years; lacked gender information; or died or had ischemic heart disease, stroke, heart failure, renal failure, bleeding esophageal varices, hepatic encephalopathy, ascites, jaundice, hepatocellular carcinoma (HCC), or hepatic failure before the index date or within 6 months after the index date. To exclude prevalent cases, we also excluded patients diagnosed as having liver cirrhosis or T2DM during 1997–1999.

## Procedures

We defined the comorbid date as the date of simultaneous diagnosis of diabetes and liver cirrhosis. Patients who took SUs for at least 28 days after the comorbid date were defined as SU users, and those who never took SUs during the study period were defined as SU nonusers. We defined the first date of SU use as the index date. Possible confounding variables in this study were age, gender, smoking status (305.1, 649.0, and V15.82), obesity [we lumped the diagnosis of overweight, abnormal weight gain, and body mass index (BMI) 25–29 as overweight (278.02, 783.1, V85.2); obesity, BMI 30–39, obesity complicated pregnancy as obesity (278.00, V77.8, 649.1, V85.3); severe obesity, BMI $\geq$40, and bariatric surgery status for obesity as severe obesity (278.01, 649.2, V45.86, V85.4)], age at T2DM diagnosis, T2DM duration, the item and number of antihypertensive and antidiabetic drugs, statin, and aspirin. In addition, comorbidities before the index date were hypertension (401–405 and A26), dyslipidemia (272, 278, A189, and A182), chronic kidney disease (CKD) (403.01, 403.11, 403.91, 404.02, 404.03, 404.12, 404.13, 404.92, 404.93, 581–588 and A350), chronic obstructive pulmonary disease (COPD; 491, 492, and 496), hepatitis C virus (HCV) infection (070.41, 070.44, 070.51, 070.54, 070.70, 070.71, and V02.62), hepatitis B virus (HBV) infection (070.2, 070.3, and V02.61), and Charlson comorbidity index (CCI) scores [12]. To evaluate the severity of diabetes, we calculated the Diabetes Complication Severity Index (DCSI) score [13].

## Main outcomes

We assessed the risks of all-cause mortality, HCC, decompensated cirrhosis, hepatic failure, major adverse cardiovascular events (MACE), and hypoglycemia. Mortality was defined as discharge from hospital with a death certificate (discharge date was defined as the death date) or termination of NHI coverage after discharge from hospital because of a critical illness and no further healthcare use for more than 1 year (the end of NHI coverage was defined as the death date). To evaluate cardiovascular and liver-related complications, we calculated the incidence of MACE, including ischemic heart disease (410–414), stroke (430–437), and heart failure (428); HCC (155.x); decompensated cirrhosis (composite of variceal bleeding, ascites, hepatic encephalopathy, and jaundice); bleeding esophageal varices; ascites; hepatic encephalopathy; and hepatic failure (570, 572.2, 572.4, or 572.8). We also investigated the incidence of emergency department visits or hypoglycemia-related admissions (251.0x, 251.1x, or 251.2x) to evaluate possible complications of treatments.

## Statistical analysis

We used propensity score matching to optimize comparability between SU users and nonusers [14]. The propensity score was estimated for each patient through nonparsimonious

multivariable logistic regression, with SU use being the dependent variable. We used 32 clinically related variables as independent covariates in the analysis (Table 1). In addition, we adopted a nearest-neighbor algorithm to construct matching pairs under the assumption that a proportion of 0.995 to 1.0 was perfect.

The incidence rate (IR) was calculated as the number of outcomes identified during the follow-up period, and divided by the total follow-up person-years for each group. Crude and multivariate-adjusted Cox proportional hazard models with robust sandwich standard error estimates were also used to compare the outcomes between SU users and nonuser. The corresponding results are presented as hazard ratios (HRs) with 95% confidence intervals (CIs). To assess the risk of all-cause mortality, we analyzed patients' dates of death or the end of the study. For other investigated outcomes, we analyzed the dates of respective outcomes or the end of follow-up on December 31, 2013, whichever came first. We compared the cumulative incidence of all-cause mortality and MACE over time between SU users and nonusers by using the Kaplan–Meier method and tested the curve difference using log-rank tests.

To evaluate the dose effect, we analyzed the risks of all-cause mortality and MACE by using three mean cumulative defined daily doses (cDDD) of SUs (<30, 30–50, and >50 cDDD/month) relative to non-SU use. The DDD for a drug represents the assumed average maintenance dose for the drug when used for its main indication in adults, which is 10 mg (= 1 DDD) for glibenclamide, 0.375 g for chlorpropamide, 1.5 g for tolbutamide, 0.5g for tolazamide, 10 mg for glipizide, 60 mg for gliquidone, 60 mg for gliclazide, and 2 mg for glimepiride. In stratified analysis, we classified SU users into three subgroups based on the first SU use: Glibenclamide, Glipizide/ Gliclazide and Glimepiride.

We set significance at a two-tailed p value of<0.05. All statistical analyses were performed using SAS v9.4 (SAS Institute, Inc., Cary, NC, USA).

## Results

### Baseline characteristics of the study participants

A total of 25,742 patients were diagnosed as having T2DM and compensated cirrhosis between January 1, 2000, and December 31, 2012. After excluding unsuitable cases, we included 7915 patients who received SU treatment for at least 28 days and 17,827 patients who never received SU treatment during the follow-up period. Fig 1 presents the study flowchart.

After propensity score matching, we selected 3,781 paired SU users and nonusers, who were similar in all variables (Table 1). Among the SU users, 20.73%, 0.1%, 0.36%, 10.96%, 0.42%, 29.15%, and 38.27% of them used glibenclamide, chlorpropamide, tolazamide, glipizide, gliquidone, gliclazide, and glimepiride, respectively. The mean age in this cohort was 57.11 years, mean duration of diabetes was 2.82 years, HBV infection rate was 20.94%, and HCV infection rate was 16.10%. The mean follow-up period was 5.55 years for SU users and 5.93 years for nonusers.

### Main outcomes

In the matched cohort of patients with T2DM and compensated liver cirrhosis, 680 (17.98%) SU users and 918 (24.28%) SU nonusers died during the follow-up period (incidence rate: 3.24 vs 4.09 per 100 patient-years). The multivariable-adjusted HR (95% CI) for SU users relative to SU nonusers was 0.79 (0.71–0.88; Table 2).

As indicated in Table 2, SU users were associated with lower risks of MACE (aHR [95% CI] 0.69 [0.61–0.80]), stroke (aHR [95% CI] 0.66 [0.53–0.83]), ischemic heart disease (aHR [95% CI] 0.66 [0.53–0.83]), and heart failure (aHR [95% CI] 0.71 [0.55–0.92]) compared with SU nonusers. Moreover, compared with SU nonusers, SU users were at a non-significantly lower

**Table 1. Baseline characteristics of sulfonylurea users and nonusers in patients with type 2 diabetes mellitus and compensated liver cirrhosis.**

| Variables | Before propensity score match | | | | After propensity score match | | | | |
|---|---|---|---|---|---|---|---|---|---|
| | Non-sulfonylurea users (n = 7552) | | Sulfonylurea users (n = 4526) | | *p* value | Non-sulfonylurea users (n = 3781) | | Sulfonylurea users (n = 3781) | | *p* value |
| | n | % | n | % | | n | % | n | % | |
| Age | | | | | < .0001 | | | | | 0.56 |
| 18–49 | 2280 | 30.19 | 1161 | 25.65 | | 1052 | 27.82 | 1012 | 26.77 | |
| 50–65 | 3366 | 44.57 | 2187 | 48.32 | | 1814 | 47.98 | 1851 | 48.96 | |
| >65 | 1906 | 25.24 | 1178 | 26.03 | | 915 | 24.20 | 918 | 24.28 | |
| Mean±SD | 56.44±11.31 | | 57.41±10.64 | | < .0001 | 57.27±11.24 | | 56.94±10.55 | | 0.20 |
| Sex | | | | | | | | | | 0.82 |
| Female | 2269 | 30.05 | 1524 | 33.67 | | 1238 | 32.74 | 1229 | 32.50 | |
| Male | 5283 | 69.95 | 3002 | 66.33 | | 2543 | 67.26 | 2552 | 67.50 | |
| DM age, mean±SD | 52.31±11.30 | | 54.93±10.64 | | < .0001 | 54.43±11.20 | | 54.17±10.47 | | 0.29 |
| DM duration, mean±SD | 4.13±3.17 | | 2.48±2.37 | | < .0001 | 2.87±2.37 | | 2.77±2.45 | | 0.08 |
| Antihypertensive drugs | | | | | | | | | | |
| ACEI/ARB | 3733 | 49.43 | 2288 | 50.55 | 0.23 | 1914 | 50.62 | 1874 | 49.56 | 0.36 |
| β-blockers | 4452 | 58.95 | 27.11 | 59.90 | 0.31 | 2265 | 59.90 | 2229 | 58.95 | 0.40 |
| Calcium-channel blockers | 2651 | 35.10 | 1644 | 36.32 | 0.17 | 1340 | 35.44 | 1332 | 35.23 | 0.85 |
| Diuretics | 2353 | 31.16 | 1462 | 32.30 | 0.18 | 1204 | 31.84 | 1178 | 31.16 | 0.52 |
| Other anti-hypertensive agent | 1805 | 23.90 | 1094 | 24.17 | 0.73 | 893 | 23.62 | 898 | 23.75 | 0.89 |
| Number of hypertensive agents | | | | | 0.01 | | | | | 0.50 |
| ≦1 | 3199 | 41.57 | 1840 | 40.65 | | 1554 | 41.10 | 1569 | 41.50 | |
| 2 | 1611 | 21.33 | 897 | 19.82 | | 799 | 21.13 | 758 | 20.05 | |
| ≧3 | 2802 | 37.10 | 1789 | 39.53 | | 1428 | 37.77 | 1454 | 38.46 | |
| Antidiabetic drugs | | | | | | | | | | |
| Metformin | 3242 | 42.93 | 2457 | 54.29 | < .0001 | 1871 | 49.48 | 1828 | 48.35 | 0.32 |
| Meglitinide | 1120 | 14.83 | 739 | 16.33 | 0.02 | 570 | 15.08 | 574 | 15.18 | 0.90 |
| Thiazolidinedione | 1171 | 15.51 | 790 | 17.45 | 0.004 | 625 | 16.53 | 601 | 15.90 | 0.45 |
| α-glucosidase inhibitor | 1135 | 13.06 | 750 | 16.57 | 0.02 | 567 | 15.00 | 567 | 15.00 | 1.00 |
| DPP-4 inhibitors | 303 | 4.01 | 254 | 5.61 | < .0001 | 170 | 4.50 | 171 | 4.52 | 0.96 |
| Insulin | 1962 | 25.98 | 1256 | 27.75 | 0.03 | 1006 | 26.61 | 988 | 26.13 | 0.64 |
| Number of oral antidiabetic drugs | | | | | < .0001 | | | | | 0.12 |
| ≦1 | 5431 | 71.91 | 3057 | 67.54 | | 2629 | 69.53 | 2664 | 70.46 | |
| 2 | 940 | 12.45 | 682 | 15.07 | | 498 | 13.17 | 555 | 14.68 | |
| 3 | 483 | 6.40 | 308 | 6.81 | | 274 | 7.25 | 230 | 6.08 | |
| ≧4 | 698 | 9.24 | 479 | 10.58 | | 380 | 10.05 | 332 | 8.78 | |
| Other drugs | | | | | | | | | | |
| Statin | 2391 | 31.66 | 1540 | 34.03 | 0.007 | 1263 | 33.40 | 1213 | 32.08 | 0.22 |
| Aspirin | 4713 | 62.41 | 2917 | 64.45 | 0.02 | 2436 | 64.43 | 2400 | 63.48 | 0.39 |
| DCSI score | | | | | < .0001 | | | | | 0.37 |
| 0 | 3263 | 43.21 | 1617 | 35.73 | | 1447 | 38.27 | 1505 | 39.80 | |
| 1 | 1376 | 18.22 | 929 | 20.53 | | 767 | 20.29 | 758 | 20.05 | |
| ≥2 | 2913 | 38.57 | 1980 | 43.75 | | 1567 | 41.44 | 1518 | 40.15 | |
| CCI index | | | | | < .0001 | | | | | 0.56 |
| 0 | 3959 | 52.42 | 2594 | 57.31 | | 2126 | 56.23 | 2160 | 57.13 | |
| 1 | 1348 | 17.85 | 773 | 17.08 | | 665 | 17.59 | 631 | 16.69 | |

(*Continued*)

**Table 1.** (Continued)

| Variables | Before propensity score match | | | | After propensity score match | | | | |
|---|---|---|---|---|---|---|---|---|---|
| | Non-sulfonylurea users (n = 7552) | | Sulfonylurea users (n = 4526) | | *p* value | Non-sulfonylurea users (n = 3781) | | Sulfonylurea users (n = 3781) | | *p* value |
| | n | % | n | % | | n | % | n | % | |
| ≥2 | 2245 | 29.73 | 1159 | 25.61 | | 990 | 26.18 | 990 | 26.18 | |
| Obesity | | | | | | | | | | |
| Overweight | 19 | 0.25 | 14 | 0.31 | 0.56 | 11 | 0.29 | 10 | 0.26 | 0.83 |
| Obesity | 185 | 2.45 | 129 | 2.85 | 0.18 | 103 | 2.72 | 100 | 2.64 | 0.83 |
| Severe obesity | 31 | 0.43 | 24 | 0.53 | 0.34 | 17 | 0.45 | 15 | 0.40 | 0.72 |
| Smoking | 427 | 5.65 | 257 | 5.68 | 0.95 | 207 | 5.47 | 214 | 5.66 | 0.73 |
| Comorbidity | | | | | | | | | | |
| Hypertension | 3043 | 40.29 | 1937 | 42.80 | 0.007 | 1590 | 42.05 | 1559 | 41.23 | 0.47 |
| Dyslipidemia | 2693 | 35.66 | 1776 | 39.24 | < .0001 | 1444 | 38.19 | 1414 | 37.40 | 0.48 |
| CKD | 1259 | 16.67 | 744 | 16.44 | 0.74 | 630 | 16.66 | 617 | 16.32 | 0.69 |
| COPD | 1504 | 19.92 | 877 | 19.38 | 0.47 | 743 | 19.65 | 720 | 19.07 | 0.50 |
| HBV | 1614 | 21.37 | 1000 | 22.09 | 0.35 | 778 | 20.58 | 805 | 21.29 | 0.44 |
| HCV | 1182 | 15.65 | 763 | 18.96 | 0.08 | 617 | 16.32 | 600 | 15.87 | 0.59 |

Abbreviations: DM, diabetes mellitus; ACEI, angiotensin converting enzyme inhibitor; ARB, angiotensin receptor blocker; CCI, Charlson comorbidity index; DCSI score, diabetes complications severity index score; CKD, chronic kidney disease; COPD, chronic obstructive pulmonary disease; HBV, hepatitis B virus; HCV, hepatitis C virus.

[a]: t-test.

risk of decompensated cirrhosis (aHR [95% CI] 0.82 [0.66–1.03], p = 0.08). We observed no significant differences between SU users and nonusers in terms of the risk of bleeding esophageal varices, ascites, hepatic encephalopathy, jaundice, hepatic failure, or HCC. SU users also exhibited no significant difference from SU nonusers in terms of the risk of severe hypoglycemia (aHR [95% CI] 0.78 [0.52–1.10]; Table 2).

Fig 2 illustrates the cumulative incidence of all-cause mortality and MACE in SU users and nonusers in patients with T2DM and compensated liver cirrhosis. Log-rank test results revealed that compared with SU nonusers, SU users were associated with lower risks of all-cause mortality and MACE.

## Dose–response relationship

Table 3 presents a comparison of dose-related all-cause mortality and MACE between SU users and nonusers. Patients who received <30, 30–50, and >50 cDDD/month of SUs exhibited a low risk of all-cause mortality compared with SU nonusers, with the corresponding aHRs (95% CI) being 0.90 (0.70–1.02), 0.82 (0.71–0.95), and 0.75 (0.65–0.88), respectively (*p* for trend: 0.004). Similarly, compared with SU nonusers, patients who received <30, 30–50, and >50 cDDD/month of SUs had a low risk of MACE, with the corresponding aHRs (95% CI) being 0.81 (0.70–0.94), 0.80 (0.69–0.94), and 0.77 (0.67–0.89), respectively (*p* for trend < 0.0001).

## Stratified analysis

S1 Table shows the results of the stratified analysis of all-cause mortality between SU users and nonusers. SU users exhibited a significantly lower risk of all-cause mortality than did SU

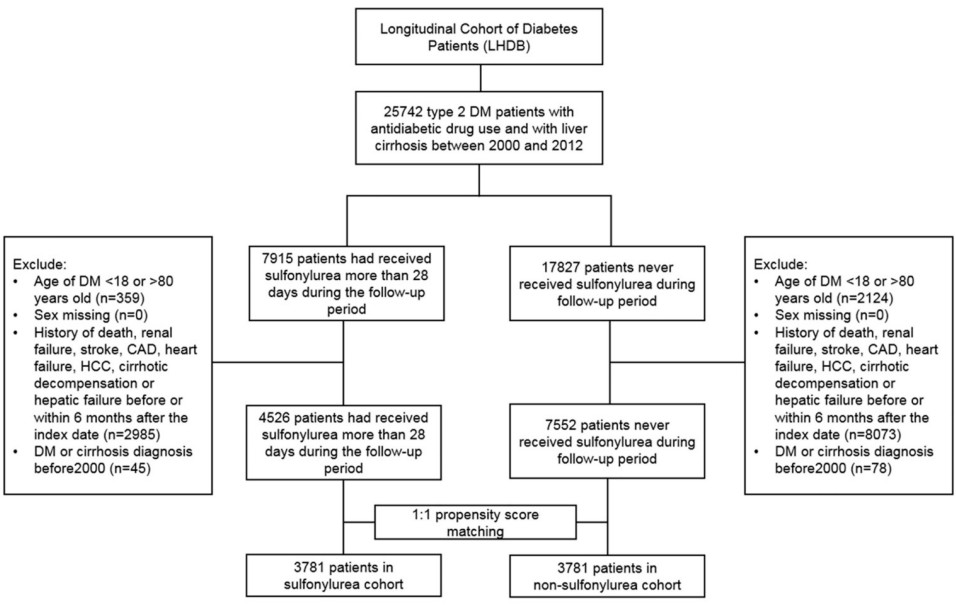

**Fig 1. The flowchart of this study.**

nonusers, except the SU users who were aged >65 years; were women; used calcium-channel blockers, thiazolidinediones, α-glucosidase inhibitor, DPP-4 inhibitors; had DCSI scores of ≥1 or CCI scores ≧2; had smoking, hypertension, CKD, or viral hepatitis infection. Compared with SU nonusers, users of glibenclamide (aHR 0.84), and glipizide/ gliclazide (aHR 0.77),

**Table 2. Outcomes of sulfonylurea users versus matched sulfonylurea nonusers in patients with type 2 diabetes mellitus and compensated liver cirrhosis.**

| Outcomes | Non-sulfonylurea users (n = 3781) | | | Sulfonylurea users (n = 3781) | | | Crude HR (95% CI) | p value | Adjusted HR (95% CI)[a] | p value |
|---|---|---|---|---|---|---|---|---|---|---|
| | Events | PY | IR | Events | PY | IR | | | | |
| All-cause mortality | 918 | 22420 | 4.09 | 680 | 20984 | 3.24 | 0.83(0.75–0.92) | < .001 | 0.79(0.71–0.88) | < .001 |
| HCC | 824 | 20418 | 4.04 | 738 | 19470 | 3.79 | 0.94(0.85–1.04) | 0.22 | 0.99 (0.90–1.11) | 0.90 |
| MACE | 559 | 20424 | 2.74 | 363 | 19946 | 1.82 | 0.66(0.58–0.76) | < .001 | 0.69(0.61–0.80) | < .001 |
| Stroke | 314 | 21453 | 1.46 | 190 | 20507 | 0.93 | 0.63(0.53–0.76) | < .001 | 0.66(0.53–0.83) | 0.003 |
| Ischemic heart disease | 219 | 21513 | 1.02 | 127 | 20538 | 0.62 | 0.60(0.48–0.75) | < .001 | 0.66(0.53–0.83) | < .001 |
| Heart failure | 170 | 21930 | 0.78 | 104 | 20730 | 0.50 | 0.65(0.51–0.84) | 0.001 | 0.71(0.55–0.92) | 0.008 |
| Decompensated cirrhosis | 795 | 20991 | 3.79 | 588 | 20011 | 2.94 | 0.77(0.70–0.86) | < .001 | 0.82(0.66–1.03) | 0.08 |
| Variceal bleeding | 44 | 22296 | 0.20 | 43 | 20897 | 0.21 | 1.03(0.68–1.58) | 0.87 | 1.09(0.56–1.93) | 0.71 |
| Hepatic ascites | 529 | 21568 | 2.45 | 414 | 20320 | 2.04 | 0.83(0.73–0.95) | < .001 | 0.83(0.63–1.09) | 0.17 |
| Hepatic encephalopathy | 464 | 21730 | 2.14 | 335 | 20553 | 1.63 | 0.77(0.66–0.88) | < .001 | 0.89(0.66–1.20) | 0.46 |
| Jaundice | 98 | 22208 | 0.44 | 62 | 20862 | 0.30 | 0.67(0.49–0.92) | 0.08 | 0.73(0.38–1.39) | 0.34 |
| Hepatic failure | 648 | 21433 | 3.02 | 480 | 20448 | 2.35 | 0.78(0.69–0.87) | < .001 | 0.83(0.65–1.05) | 0.12 |
| Hypoglycemia | 103 | 22147 | 0.47 | 70 | 20793 | 0.34 | 0.74(0.54–1.00) | 0.05 | 0.78(0.52–1.10) | 0.10 |

Abbreviations: PY, person-years; IR, incidence rate, per 100 person-years; HR, hazard ratio; CI, confidence interval; DM, diabetes mellitus; HCC, hepatocellular carcinoma; MACE, major adverse cardiac event, including stroke, ischemic heart disease, and heart failure.

[a]Adjusted for age, sex, index year, age at DM diagnosis, DM duration (years), antihypertensive drugs (ACE inhibitors, ARBs, β-blockers, calcium-channel blockers, diuretics, other antihypertensive), anti-diabetic drugs (metformin, meglitinides, thiazolidinedione, α-glucosidase inhibitor, DPP-4 inhibitors, insulin), statin, aspirin, CCI index (0, 1, ≥2), DCSI score (0, 1, ≥2), obesity, smoking, hypertension, dyslipidemia, CKD, COPD, HBV, and HCV.

**(A)**

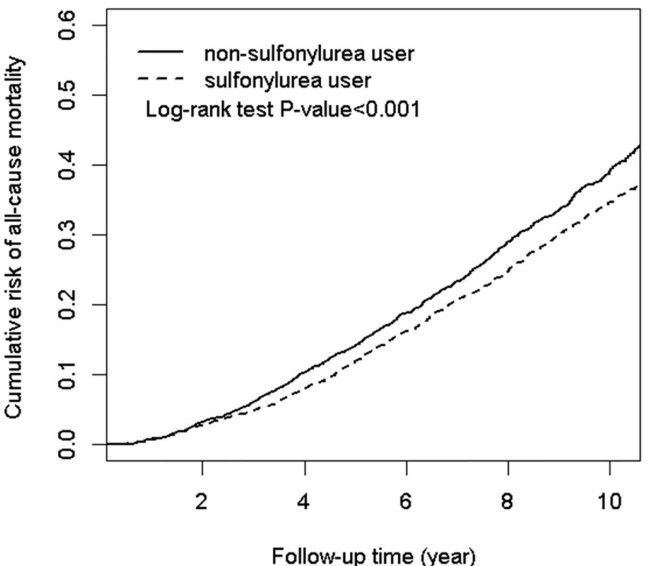

**(B)**

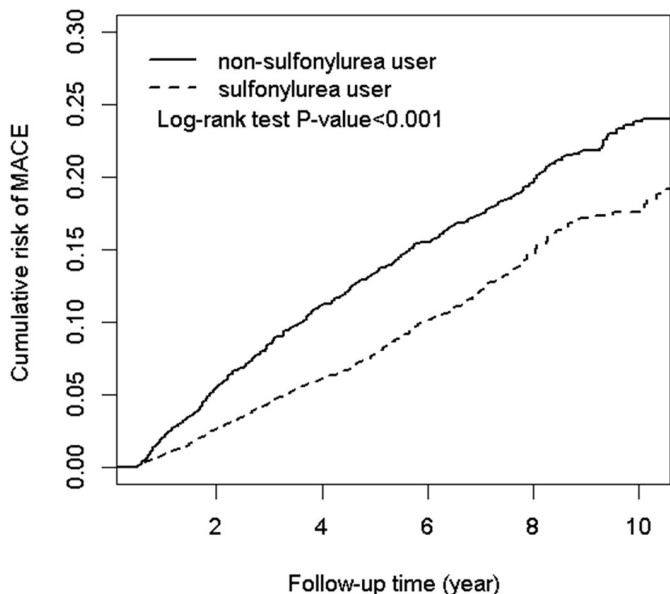

**Fig 2. Cumulative incidence curves of (A) all-cause mortality and (B) major adverse cardiovascular events (MACE) between sulfonylurea users and nonusers in patients with diabetes and compensated cirrhosis.**

**Table 3. Hazard ratios and 95% confidence intervals for all-cause mortality and cardiovascular events associated with cumulative average dose of sulfonylureas.**

| Variable | All-cause mortality | | | | Crude HR (95%CI) | Adjusted HR (95%CI)[a] |
|---|---|---|---|---|---|---|
| | n | Event | PY | IR | | |
| Sulfonylurea dose (DDD per months) | | | | | | |
| Non-users | 3781 | 918 | 22420 | 4.09 | 1 (reference) | 1 (reference) |
| <30 | 1520 | 234 | 7139 | 3.28 | 0.92(0.79–1.06) | 0.90(0.70–1.02) |
| 30–50 | 1067 | 216 | 6402 | 3.37 | 0.84(0.72–0.97)* | 0.82(0.71–0.95)** |
| >50 | 1194 | 230 | 7442 | 3.09 | 0.75(0.65–0.86)*** | 0.75(0.65–0.88)*** |
| P for trend | | | | | | 0.004 |
| Variable | Major adverse cardiac events | | | | Crude HR (95%CI) | Adjusted HR (95%CI)[a] |
| | n | Event | PY | IR | | |
| Sulfonylurea dose (DDD per months) | | | | | | |
| Non-users | 3781 | 559 | 20424 | 2.74 | 1 (reference) | 1 (reference) |
| <30 | 1531 | 124 | 6861 | 1.81 | 0.67(0.55–0.81)*** | 0.81(0.70–0.94)** |
| 30–50 | 1058 | 110 | 6016 | 1.83 | 0.66(0.54–0.81)*** | 0.80(0.69–0.94)** |
| >50 | 1192 | 129 | 7068 | 1.83 | 0.66(0.55–0.80)*** | 0.77(0.67–0.89)*** |
| P for trend | | | | | | < .0001 |

*$p < 0.05$,

**$p < 0.01$,

***$p < 0.001$.

PY, person-years; IR, incidence rate, per 100 person-years; HR, hazard ratio; CI, confidence interval; DM, diabetes mellitus; DDD, defined daily doses.

[a]Adjusted for age, sex, index year, age at DM diagnosis, DM duration (years), antihypertensive drugs (ACE inhibitors, ARBs, β-blockers, calcium-channel blockers, diuretics, other antihypertensive), antidiabetic drugs (metformin, meglitinides, thiazolidinedione, α-glucosidase inhibitor, DPP-4 inhibitors, insulin), statin, aspirin, CCI index (0, 1, ≥2), DCSI score (0, 1, ≥2), obesity, smoking, hypertension, dyslipidemia, CKD, COPD, HBV, and HCV.

glimepiride (aHR 0.67) seemed to be associated with a lower HR of mortality. S2 Table presents the results of stratified analysis of MACE between SU users and nonusers. SU users exhibited a significantly lower risk of MACE than did SU nonusers, except those SU users who were aged >65 years; used diuretics, meglitinide, thiazolidinediones, α-glucosidase inhibitor, or DPP-4 inhibitors; had smoking or viral hepatitis infection. Glimepiride use (aHR 0.55) seemed to be associated with a lower HR of cardiovascular events compared with glibenclamide use (aHR 0.73), and glipizide/gliclazide use (aHR 0.71).

## Discussion

After propensity score matching, this nationwide cohort study demonstrated that SU use was associated with lower risks of all-cause mortality and major cardiovascular events compared with SU non-use in patients with T2DM and compensated cirrhosis. The lower risks of death and cardiovascular events observed among SU users suggest a dose–response relationship. Glimepiride seemed to be associated with lower risks of death and cardiovascular diseases compared with glibenclamide, glipizide and gliclazide in this study.

The University Group Diabetes project trial in 1970 reported that SUs showed cardiovascular (CV) risks [15]. Nevertheless, the UK Prospective Diabetes Study (UKPDS) compared intensive diabetes treatment using SUs or insulin with conventional (diet) control; the study demonstrated that the use of SUs does not carry a high risk of CV disease [16]. A subsequent observational study of the UKPDS demonstrated that SU use showed reduced macrovascular complications with improved glycemic control [16]. Some randomized clinical studies [17–19] comparing SUs with other hypoglycemic agents have demonstrated that SU use is not associated with increased CV risk. Our study revealed that SU users were associated with a lower

risk of MACE than did SU nonusers. The inconsistency between our findings and those of the aforementioned randomized studies may be that our study included people with simultaneous T2DM and liver cirrhosis, who were not observed before. The lower CV risk associated with SU use in this study may be attributed to the glucose-lowering effect of SUs, as indicated in the UKPDS [16] and other studies on glucose-lowering effects [20].

Meta-analyses have demonstrated no consistent association between all-cause mortality and SU use in patients with T2DM. Some studies have reported SU use to be associated with increased mortality risk [21, 22], whereas others have reported SU use to not be associated with increased mortality risk [23]. Our study revealed that SU use was associated with a lower risk of all-cause mortality compared with SU nonuse; this SU-associated lower risk of mortality suggests a dose–response relationship. We observed that SU users had a lower risk of CV disease, potentially lower risk of cirrhotic decompensation, and no increased risk of hypoglycemia compared with SU nonusers; these three factors may explain the SU-associated lower mortality risk in this study.

Our study also suggested that glimepiride is associated with lower risks of CV events and all-cause mortality compared with glipizide, gliclazide, and glibenclamide. Possible explanations for this finding are as follows: (1) Glimepiride binds less avidly with cardiac tissue and could maintain ischemic myocardial preconditioning; however, glibenclamide appears to inhibit it [24]. (2) The elimination and pharmacokinetics of glimepiride are not changed in patients with renal insufficiency and significant liver disease, respectively [25]. (3) Glimepiride has a lower association with hypoglycemia than does glibenclamide [7]. (4) Glimepiride improves hyperinsulinemia and atherosclerosis through extra pancreatic effects [26]. (5) Glimepiride is associated with a lower mortality risk compared with glibenclamide, as indicated by a network meta-analysis [27].

Studies have reported that T2DM aggravates the progression of liver cirrhosis [4]. However, few studies have detailed the association between SU use and cirrhosis progression. Our results reveal that SUs seem to have the tendency of attenuating cirrhotic decompensation. The reason for this is currently unclear. In a study using cirrhotic rats, glibenclamide significantly increased portal and systemic vascular resistance initially and then decreased portal pressure and increased systemic vascular resistance [28].

Because most SUs are metabolized in the liver and cleared by the kidney, caution should be exercised when prescribing SUs to patients with advanced liver disease [7]. Large clinical studies on the safety of SU use in liver cirrhosis are unavailable [29]. SU-induced hepatotoxicity has rarely been reported for glycemic management in patients with T2DM [29]. Nevertheless, our study reported no association between high risk of hepatic failure and SU use in patients with T2DM and compensated liver cirrhosis.

HCC is frequently observed in patients with liver cirrhosis; diabetes can increase HCC risk [4]. Previous research reported the association of SU use and HCC development [30]. SUs can increase pancreatic β cells to secrete insulin, and insulin is a growth-promoting hormone with mitogenic effects. However, SU use in patients with liver cirrhosis was not associated with a relatively high risk of HCC in our study. The reason for this inconsistency between our study and previous studies may be attributed to the different study populations.

The most concerning side effect of SU use is hypoglycemia. Large clinical studies have reported the incidence of severe hypoglycemia to be 0.2 to 0.4 per 1000 person-years [1]. Patients with renal or hepatic impairment or elderly people are particularly vulnerable to hypoglycemia [1, 7]. Patients with liver cirrhosis may have dysfunction in gluconeogenesis and shortage of glycogen storage, reduced metabolism of antidiabetic drugs, impaired glucagon catabolism and increased risk of hypoglycemia [31]. However, our study revealed that SU use in persons with compensated liver cirrhosis was not associated with a higher risk of severe hypoglycemia compared with SU non-use.

Our study has some limitations that should be addressed. First, our administrative claims dataset has no information on alcohol use, physical activity, or family history. It also has no information on blood biochemical and hemoglobin A1C results. Therefore, we can't calculate the Child-Pugh scores and albumin-bilirubin (ALBI) grade to evaluate the severity of liver dysfunction [32] and the treatment situation of T2DM. These unmeasured variables may influence our outcomes. However, we matched the item and number of antihypertensive and oral antidiabetic drugs to decrease the difference of blood pressure and glycemic control, used clinical diagnoses to distinguish patients into compensated and decompensated liver cirrhosis groups, and used DCSI and diabetes duration to separate patients as per the severity of T2DM, Moreover, we performed propensity score matching to adjust major variables between SU users and nonusers to maximally reduce the bias and confounding factors. Second, the patients' adherence to the regimen of hypoglycemic agents cannot be appropriately assessed in this health insurance database. The doctors' and patients' preference for particular hypoglycemic agents may also influence our variables. Finally, because this was a Chinese population-based cohort study, the results may not be applicable to other ethnicities.

## Conclusions

Patients with liver cirrhosis are prone to developing diabetes. Because of the lack of randomized clinical trials, a large cohort study may provide valuable clinical information on the association of SU use and mortality risk. Our study demonstrated that SU use in patients with T2DM and compensated cirrhosis was associated with lower risks of all-cause mortality and major cardiovascular events. Therefore, SUs may be useful forT2DM management in patients with liver cirrhosis.

## Supporting information

**S1 Table. Stratified analysis of all-cause mortality associated with sulfonylurea use and nonuse.**
(DOCX)

**S2 Table. Stratified analysis of major adverse cardiovascular events associated with sulfonylurea use and nonuse.**
(DOCX)

## Acknowledgments

This manuscript was edited by Wallace Academic Editing.

## Author Contributions

**Conceptualization:** Fu-Shun Yen, James Cheng-Chung Wei, Chii-Min Hwu, Chih-Cheng Hsu.

**Data curation:** Lu-Ting Chiu, Chii-Min Hwu, Chih-Cheng Hsu.

**Formal analysis:** Jung-Nien Lai.

**Funding acquisition:** Jung-Nien Lai, Lu-Ting Chiu, Ming-Chih Hou.

**Supervision:** Ming-Chih Hou, Chih-Cheng Hsu.

**Writing – original draft:** Fu-Shun Yen, Lu-Ting Chiu, Chih-Cheng Hsu.

**Writing – review & editing:** Fu-Shun Yen, Ming-Chih Hou.

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
