## [Decision Letter · Decision Letter 0]

23 Oct 2020

PONE-D-20-31507

Sulfonylureas may be useful for glycemic management in patients with diabetes and liver cirrhosis

PLOS ONE

Dear Dr. Chih-Cheng Hsu,

Thank you for submitting your manuscript to PLOS ONE. After careful consideration, we feel that it has merit but does not fully meet PLOS ONE’s publication criteria as it currently stands. Therefore, we invite you to submit a revised version of the manuscript that addresses the points raised during the review process.

We look forward to receiving your revised manuscript.

Kind regards,

Tatsuo Kanda, M.D., Ph.D.

Academic Editor

PLOS ONE

Journal Requirements:

Reviewers' comments:

Reviewer's Responses to Questions

**Comments to the Author**

1. Is the manuscript technically sound, and do the data support the conclusions?

Reviewer #1: Yes

Reviewer #2: No

2. Has the statistical analysis been performed appropriately and rigorously? 

Reviewer #1: Yes

Reviewer #2: I Don't Know

3. Have the authors made all data underlying the findings in their manuscript fully available?

Reviewer #1: Yes

Reviewer #2: Yes

4. Is the manuscript presented in an intelligible fashion and written in standard English?

Reviewer #1: Yes

Reviewer #2: Yes

5. Review Comments to the Author

Reviewer #1: Authors reported SU use in patients with T2DM and compensated cirrhosis was associated with lower risks of all-cause mortality and major cardiovascular events. There are no major revisions required. Minor revisions required are as follows:

#1: Full spelling is required only for the first appearance of the abbreviation. Please correct the 113th, 123th, 129th, 131st, 132nd, 133rd, 137th, 192nd, 193rd, 226th, 232nd, 247th, 272nd, and 273rd line.

#2: Page 17, line 3: the correct one is ‘‘HCC’’, not ‘‘Hepatocellular’’.

Reviewer #2: #1 The authors defined decompensated liver cirrhosis as LC with variceal bleeding, ascites, encephalopathy, or jaundice. Under this definition, patients with compensated LC includes wide ranges of liver function. The information of liver function such as Child-Pugh score or ALBI grade is necessary.

#2 Patients with liver cirrhosis has a risk of hypoglycemia due to dysfunction in gluconeogenesis and shortage of glycogen. The authors concluded that the usage of SUs was not associated with sever hypoglycemia. The authors should mention the risk of the hypoglycemia due to LC itself in the Discusion.

#3 Space may be necessary in line 89 Page 6 (between in and 2000), line 182 Page 11 (between 7915 and patients).

6. PLOS authors have the option to publish the peer review history of their article (what does this mean?). If published, this will include your full peer review and any attached files.

Reviewer #1: **Yes: **Hidehiro Kamezaki

Reviewer #2: No

---

## [Author Response · Author response to Decision Letter 0]

15 Nov 2020

Tatsuo Kanda, M.D., Ph.D.

Academic Editor

PLOS ONE 

November 2, 2020

Dear Prof. Tatsuo Kanda:

Re: Document reference No. PONE-D-20-31507

Please find attached a revised version of our document “Sulfonylureas may be useful for glycemic management in patients with diabetes and liver cirrhosis”. We would like to resubmit for publication as an original article in PLOS ONE. 

Your comments and those of the reviewers were highly insightful and enabled us to improve the quality of our document. In the following pages are our responses to each comment from the reviewer(s) as well as your own comments.

Revisions in the text are shown in yellow highlights. We hope that our revisions to the document combined with our accompanying responses will be sufficient to render our document suitable for publication in PLOS ONE.

Yours sincerely,

Chih-Cheng Hsu, MD, DrPH

Institute of Population Health Sciences, National Health Research Institutes

Tel.: +886 37 246166 #36336

Fax: +886 37 586463

E-Mail: cch@nhri.edu.tw

Address: 35 Keyan Road, Zhunan, Miaoli County 35053, Taiwan

Responses to the comments of Editor

Response: Thank you for your encouragement! We will comply with the style requirement of PLOS ONE. 

2. We note that you have indicated that data from this study are available upon request. PLOS only allows data to be available upon request if there are legal or ethical restrictions on sharing data publicly. For information on unacceptable data access restrictions, please see http://journals.plos.org/plosone/s/data-availability#loc-unacceptable-data-access-restrictions. In your revised cover letter, please address the following prompts:

Response: Data of this study are available from the National Health Insurance Research Database (NHIRD) published by Taiwan National Health Insurance (NHI) Administration. The data utilized in this study cannot be made available in the paper, the supplemental files, or in a public repository due to the ‘‘Personal Information Protection Act’’ executed by Taiwan government starting from 2012. Requests for data can be sent as a formal proposal to the NHIRD Office (https://dep.mohw.gov.tw/DOS/cp-2516-3591-113.html) or by email to stsung@mohw.gov.tw. 

Response: Yes, we have ORCID iDs and that they are validated in Editorial Manager.

Response: Thanks! We have deleted the ethics statement written in other section besides the Methods. 

Responses to the comments of Reviewer #1

1. Authors reported SU use in patients with T2DM and compensated cirrhosis was associated with lower risks of all-cause mortality and major cardiovascular events. There are no major revisions required. Minor revisions required are as follows:

#1: Full spelling is required only for the first appearance of the abbreviation. Please correct the 113th, 123th, 129th, 131st, 132nd, 133rd, 137th, 192nd, 193rd, 226th, 232nd, 247th, 272nd, and 273rd line.

Response: Thank you for careful reading of our manuscript! We have corrected the wrong spellings in these areas. 

2. Page 17, line 3: the correct one is ‘‘HCC’’, not ‘‘Hepatocellular’’. 

Response: Thanks! We have corrected this error on page 17. 

Responses to the comments of Reviewer #2

1. The authors defined decompensated liver cirrhosis as LC with variceal bleeding, ascites,

encephalopathy, or jaundice. Under this definition, patients with compensated LC includes wide ranges of liver function. The information of liver function such as Child-Pugh score or ALBI grade is necessary. 

Response: We are so sorry that our administrative dataset is lack of blood biochemical test results. Therefore, we can’t calculate the Child-Pugh scores and albumin-bilirubin (ALBI) grade to evaluate the severity of liver dysfunction. We have described this limitation of our study on page 25-26 as “It also has no information on blood biochemical and hemoglobin A1C results. Therefore, we can’t calculate the Child-Pugh scores and albumin-bilirubin (ALBI) grade to evaluate the severity of liver dysfunction [32] and the treatment situation of T2DM”. 

2. Patients with liver cirrhosis has a risk of hypoglycemia due to dysfunction in gluconeogenesis and shortage of glycogen. The authors concluded that the usage of SUs was not associated with sever hypoglycemia. The authors should mention the risk of the hypoglycemia due to LC itself in the Discussion. 

Response: Thank you so much for this important suggestion. We have described the risk of hypoglycemia due to liver cirrhosis on page 25 as” Patients with liver cirrhosis may have dysfunction in gluconeogenesis and shortage of glycogen storage, reduced metabolism of antidiabetic drugs, impaired glucagon catabolism and increased risk of hypoglycemia [31]”. 

3. Space may be necessary in line 89 Page 6 (between in and 2000), line 182 Page 11 (between 7915 and patients).

Response: Thanks! We have corrected these errors.

---

## [Decision Letter · Decision Letter 1]

26 Nov 2020

Sulfonylureas may be useful for glycemic management in patients with diabetes and liver cirrhosis

PONE-D-20-31507R1

Dear Dr. Chih-Cheng Hsu,

We’re pleased to inform you that your manuscript has been judged scientifically suitable for publication and will be formally accepted for publication once it meets all outstanding technical requirements.

Kind regards,

Tatsuo Kanda, M.D., Ph.D.

Academic Editor

PLOS ONE

Additional Editor Comments (optional):

Reviewers' comments:

Reviewer's Responses to Questions

**Comments to the Author**

1. If the authors have adequately addressed your comments raised in a previous round of review and you feel that this manuscript is now acceptable for publication, you may indicate that here to bypass the “Comments to the Author” section, enter your conflict of interest statement in the “Confidential to Editor” section, and submit your "Accept" recommendation.

Reviewer #1: All comments have been addressed

Reviewer #2: All comments have been addressed

2. Is the manuscript technically sound, and do the data support the conclusions?

Reviewer #1: Yes

Reviewer #2: Yes

3. Has the statistical analysis been performed appropriately and rigorously? 

Reviewer #1: Yes

Reviewer #2: I Don't Know

4. Have the authors made all data underlying the findings in their manuscript fully available?

Reviewer #1: Yes

Reviewer #2: Yes

5. Is the manuscript presented in an intelligible fashion and written in standard English?

Reviewer #1: Yes

Reviewer #2: Yes

6. Review Comments to the Author

Reviewer #1: (No Response)

Reviewer #2: Thank you for submitting the revise. I think there are no more major revisions required in the manuscript.

7. PLOS authors have the option to publish the peer review history of their article (what does this mean?). If published, this will include your full peer review and any attached files.

Reviewer #1: **Yes: **Hidehiro Kamezaki

Reviewer #2: No

---

## [Editor Report · Acceptance letter]

3 Dec 2020

PONE-D-20-31507R1 

Sulfonylureas may be useful for glycemic management in patients with diabetes and liver cirrhosis 

Dear Dr. Hsu:

I'm pleased to inform you that your manuscript has been deemed suitable for publication in PLOS ONE. Congratulations! Your manuscript is now with our production department. 

Kind regards, 

on behalf of

Dr. Tatsuo Kanda 

Academic Editor

PLOS ONE